# Practices of herbal management of malaria among trading mothers in Shai Osudoku District, Accra

**Evans Osei Appiah**[1]*, **Stella Appiah**[2], **Ezekiel Oti-Boadi**[3], **Albert Oppong-Besse**[4], **Dorothy Baffour Awuah**[3], **Priscilla Ofosuhemaa Asiedu**[3], **Lt Emmanuel Oti-Boateng**[5]

**1** Department of Midwifery, School of Nursing and Midwifery, Valley View University, Oyibi, Ghana, **2** Head of Nursing Department, School of Nursing and Midwifery, Valley View University, Accra, Ghana, **3** Department of Nursing, School of Nursing and Midwifery, Valley View University, Accra, Ghana, **4** Nursing Department, Valley View University, Accra, Ghana, **5** 37 Military Hospital, Accra, Ghana

* oseiappiahevans@ymail.com

## Abstract

### Background

Malaria is one of the leading causes of morbidity in the world. It is a significant health concern in most developing countries, including Ghana. Even though there are several orthodox medications used for decades in treating malaria effectively, a substantial number of individuals in developing countries are resorting to the use of herbs in the treatment of malaria. The study aim at exploring the practices of herbal management of malaria among trading mothers in Shai Osudoku District, Accra.

### Methods

A qualitative approach with an exploratory, descriptive design was adopted in analyzing the research problem. Purposive sampling technique was used to select twenty (20) participants to partake in a face-face interview, guided by a semi-structured interview guide. The data were transcribed verbatim and analysed by adopting content analysis.

### Results

Two significant themes and seven subthemes were generated following the analysis of this study. The main themes were; preferences for herbal malaria treatment and the practices and effectiveness of herbal medicine used for malaria treatment. It was worth noting that the women's cultural beliefs did not influence their preference for herbal malaria treatment. The main challenge associated with the herbal malaria treatment was inappropriate dosage specification.

### Conclusion

This study discovered that several factors influenced participants' preferences for malaria treatment. Participants further listed some traditional ways of treating malaria which implies that there is herbal malaria practice. However, literature in this area is inadequate, and most

**Data Availability Statement:** All relevant data are within the manuscript.

**Funding:** The author(s) received no specific funding for this work.

**Competing interests:** The authors have declared that no competing interests exist.

**Abbreviations:** ACT, artemether-lumefantrine, Shai Osudoku District; AM-L, Artemether-lumefantrine; AS+AQ, Artesunate-amodiaquine; DHRC, Dodowa Health Research Center; GSS, Ghana Statistics Service; IRB, Institutional Review Board; ITNs, Insecticide-treated nets.

herbs lack specifications for use. It is therefore recommended that future research focus on scientific herbal malaria treatment. Also, regulating bodies should ensure that quality herbal drugs are sold for consumption.

## Introduction

Malaria is one of the leading causes of morbidity in the world [1]. It is a significant health concern in most developing countries, including Ghana. Malaria is a severe disease that causes a high fever and chills and is transmitted by an infected female Anopheles mosquito bite [2]. It is estimated that 10,000 women and 200,000 infants die yearly-43e due to malaria infection during pregnancy. Ghanaians are at increased risk for malaria due to the increased number of mosquitoes [3]. Malaria was responsible for 19% of the deaths in Ghana in 2015; malaria admissions alone increased from 280,000 to 340,000 persons between 2000 and 2017 [4]. Furthermore, out of 11,880,000 malaria cases recorded, 11,880 deaths were confirmed to be caused by malaria from 2000 to 2017 [5]. It is, therefore, necessary for more attention to be geared towards the treatment of malaria using both conventional and herbal means.

Malaria has long been treated with orthodox medications such as *chloroquine*, Artemisinin-based combination therapy, intravenous artesunate and ACT (*artemether-lumefantrine or dihydroartemisinin-piperaquine* as well as herbal treatment [6–10]. However, evidence suggests that challenges associated with medical treatment of malaria include increased resistance to malaria and inaccessible health facilities in some developing countries, which have contributed to increased utilization of herbal therapy in most developing countries such as Ghana, Kenya, Zimbabwe, and Nigeria [3, 10, 11].

Herbal medicine is used to treat malaria among all categories, including children under five years [12]. Common medicinal plants used in treating malaria are *Moringa oleifera, Sarcocophalus lalifolius and Cassia sieberiana Bambusa vulgaris Schrad. ex J.C. Wendl. (Poaceae), Deinbollia pinnata Schum &Thonn. (Sapindaceae), Elaeis guineensis Jacq. (Arecaceae), Greenwayodendron sp. (Annonaceae) and Solanum torvum Sw (Solanaceae) Morinda Lucida Benth., Indigofera sp. and Nauclea latifolia Sm* [5, 13, 14].

Malaria prevalence in Ghana is on the ascendancy, with rising in the number of babies affected [12, 15]. Even though there are several orthodox treatments for malaria in various health facilities that are effective in treating malaria, some authors in Ghana have reported that most Ghanaians prefer the herbal management of malaria because it is less costly and has fewer side effects, efficient and available [16]. Also, literature on Malaria treatment with herbal control is scarce despite the increased utilization of herbs in treating malaria, which could be attributed to the fact that most of these practices are not documented. The researchers, therefore, intend to explore the techniques of herbal management of malaria among trading mothers in Shai Osudoku District, Accra.

## Methodology

### Research design

The researchers adopted a qualitative exploratory research design for this study. Qualitative research method does not centre its interest on establishing statistical findings of the phenomenon but instead gains an in-depth understanding of people's beliefs, attitudes, and experiences. The qualitative exploratory design allows participants to voice their concerns about a specific topic and enhances participants' involvement. The target population for this study was

the mothers from Shai Osudoku District trading in their District Market centre since the mothers trading in their district market were from more than 10 rural communities in the district (Dodowa, Kordiabbe, Doryumu, Sota, Mokomeshitamohe, Kadjanya, Volivo, Agbekotsekpo, Abuvienu, Adakope, Ayikuma, Ayenya, Agomeda, Asutware, Osuwem, Tokpo, Agortor, Natriku, Kasunya, Chebitenya, and Odumse).

The inclusion criteria for this study were women: a) who lived in any of the communities in the District, who could express themselves in Twi (local dialect), or English which are the languages spoken by the researcher or Ga-Adangbe, which is the native language of the people and b) who were willing to participate in the study. The exclusion criteria of the study included: a) women who are mentally impaired.

This study adopted the purposive sampling technique known as "judgment sampling" to select participants. With this type of sampling technique, participants were deliberately chosen from the population that qualifies for the study. Purposive sampling technique was used for the analysis because all participants had the appropriate features for the research, and they also provided the information necessary for the study. The sample size for qualitative research is determined at the point of saturation [17]. Based on this, the researchers ended the data collection when no new information was obtained from the participants hence the sample size of this study based on saturation was 20.

Written and verbal consent was obtained from all the 20 participants. Ethical clearance was obtained from Dodowa Health Research Centre Institutional Review Board (DHRC-IRB) which is under Ghana health Service with the protocol identification number DHRCIRB/38/03/20 before data was collected.

Face-to-face in-depth interview with a semi-structured interview guide was conducted to determine the practices and preferences of herbal management of malaria among trading mothers. A tape recorder was used to capture the interviews during the recording. Before data collection. Permission was sought from the leaders in the market after. Further on, permission was obtained from the participants. Participants were informed about a consent form that needed to be filled out after agreeing to participate in the study. The filling of the consent form was done before the interview session. Participants' contacts were collected, and they were called later to arrange a specific date, time and venue for the interview. The discussion was intended to last for about one month about the availability of participants. At least two interviews were conducted per day, lasting 40 minutes to 1 hour per participant.

Rigour is the measure of ensuring that the research result is valid [18]. The researcher stated that rigorous research result is based trustworthiness of a study's findings. Trustworthiness is the authenticity and truthfulness of the findings of a study [19]. In addition, credibility, transferability, dependability, and confirmability are used to describe trustworthiness better. Credibility was explained as the accuracy and authenticity of a research finding [20]. This was ensured by transcribing the responses of participants verbatim. Also, the research supervisor reviewed the work severally to ensure the study was done based on the objectives.

With transferability, the researchers described in detail the methods used such as the design, sampling technique, sample size, data collection tool, and procedure to help researchers who want to conduct similar studies in other settings. Dependability and confirmability were also ensured in this study by following rigorous systematic process such as obtaining ethical clearance for this study, ensuring verbatim transcription, pretesting the data, ensuring that the whole study was based on the study objectives, and ensuring participants met the inclusion criteria were recruited.

Data analysis is the process of inspecting, modifying and transforming data to gain helpful information. Content analysis was used to analyze data. The type of content analysis employed was conventional as coding categories were derived directly from the content data. A

researcher explained that content analysis is used to organize and draw meaningful conclusions from obtained data [21]. With this, the researchers transcribe the data verbatim and read severally through the transcripts to familiarize themselves with that data and understand the participants' responses. Following this, the raw data was coded with a few words that that had exact meaning to what participants said and later generated into themes making sure the meaning is still not lost.

## Results

Twenty (20) participants with a history of malaria were interviewed. The participants were between twenty-six (26) and fifty-nine (59) years and were selected during the face-to-face interactions at the market. Fifteen (75%) were married, and five (35%) were single. The languages used during the interview were Ga, Twi, and English. For education, ten (50%) had JSS (Junior High School) level of education, 6 (30%) also had form 4 level of education, and 4 (20%) of them had SSS (Senior High School) level of education. All the participants (100%) were Christians, and for the number of children, 16 (80%) of them had more than two children, and 4 (20%) of them had a child. The details of the socio-demographic characteristics are illustrated in Table 1.

### Organization of themes

Two themes and 7 subthemes were generated from the study analysis. The details are presented in Table 2.

### Preferences for herbal malaria treatment

Mainly, the choice of treatment for a disease or illness is influenced by many factors ranging from experience with the treatment, recommendation from others, the success rate of that particular treatment and side effects. Therefore, several factors influenced the participant's choice for malaria treatment. This consisted of cultural practices to how ineffective some drugs are to some of the participants. Others choices of malaria treatment were also influenced by their family, and some were using their current therapy due to the side effects they had experienced. Under this theme, five sub-themes emerged:

### Cultural practices

Cultural practices play a significant role in Ghanaian culture and among Ghanaians. This role is so vital that most Ghanaians do not consider participating or getting involved with factors against their cultural practices. However, culturally, none of the participants had any rules or knew of any artistic tradition that had influenced how they treated malaria. However, some of the participants stated that their grandmothers treated them for malaria when they were children, and they might have gained their knowledge from their culture but could not confirm it. This was revealed in comments made by participants;

> "*I don't know of any cultural practice, but what I can say is that as I was young, I remember my grandmother using the neem tree anytime she suspected we had malaria, and she would also add bitter leaves in the treatment. She grinds the bitter leaves and adds them to the neem tree leaves for you to drink. So anytime we get a fever or feel cold, she will do that for my brothers and me. I feel she got that knowledge from her people*".

> *P16*

**Table 1. Demographic characteristics of the respondents.**

| Variable | Frequency (n = 20) | Percent (%) |
|---|---|---|
| **Age (years)** | | |
| 20–30 | 6 | 30 |
| 31–40 | 7 | 35 |
| 41–50 | 5 | 25 |
| ≥50 | 2 | 10 |
| **Place of Residence** | | |
| Bawaleshie bela | 5 | 25 |
| Seduase | 6 | 30 |
| Wodokum | 4 | 20 |
| Oyibi town | 2 | 10 |
| Dodowa | 3 | 15 |
| **Religion** | | |
| Christian | 20 | 100 |
| **Educational Level** | | |
| JSS (Junior High students) | 10 | 50 |
| SSS (Senior High Students) | 4 | 20 |
| Primary School Students | 6 | 30 |
| **Duration of Selling (years)** | | |
| 0–10 | 10 | 50 |
| 11–20 | 9 | 45 |
| ≥21 | 1 | 5 |
| **Number of Children** | | |
| One child | 4 | 20 |
| ≥ more than two children | 16 | 80 |
| **Tribe** | | |
| Ga | 8 | 40 |
| Fante | 4 | 20 |
| Ewe | 6 | 30 |
| Ga Adangbe | 2 | 10 |
| **Marital Status** | | |
| Single | 4 | 20 |
| Married | 16 | 80 |
| **Duration of Marriage (years)** | | |
| 0–10 | 10 | 50 |
| ≥11and above | 10 | 50 |
| **Income per Month** | | |
| 100–1000 | 17 | 85 |
| ≥1001and above | 3 | 15 |

Some participants did not have any idea of any cultural practices and some also preferred hospital management of malaria to herbal treatment since herbal remedies can lead to organ complications.

"*I was born and bred in Accra, so there is a lot I don't know about my culture*".

*P7*

**Table 2. Themes and subthemes.**

| THEMES | SUB-THEMES |
|---|---|
| Preference for herbal malaria treatment | 1. Cultural practices |
| | 2. Side effects of antimalarial drug |
| | 3. Ineffectiveness of antimalarial drug |
| | 4. Herbal malaria treatment practices |
| 2. Practices and effectiveness of herbal medicine use in malaria treatment. | 5. Home preparation for malaria |
| | 6. Dosage for herbal preparation |
| | 7. Effectiveness of herbal malaria treatment |

"*Well, I can't put my finger on any cultural practice that influenced my treatment choice for malaria but me, I prefer the hospital medicine because now everything we eat has been changed, so to be on the safer side, I take the one that the doctors prescribe because when you take more of the natural remedies, it can lead to kidney problems and all that.*"

*P15*

"*I don't know of any cultural things that influenced my malaria treatment. As I said, I like to go to the hospital when I feel I have malaria but what I remember is that when I was a kid, my auntie would boil water and put a naphthalene ball in it and make you sit behind it with a big cloth covering you anytime I begin to feel hot and cold dizziness and vomiting. She also did the same thing but with neem tree leaves.*"

*P4*

## Side effects of antimalarial drug

Side effects are widespread with most drugs. Apart from the therapeutic effect the drug is supposed to give, it also comes along with other products that affect the body negatively. Side effects can range from constipation or diarrhoea to dizziness or weakness. Except for just a few, most of the participants had experienced one or two adverse side effects from taking the antimalarial drug, with most of the side effects being dizziness and feeling weak. Some participants also stated that they used to have side effects of weakness when they took antimalarial drugs, but they do not currently experience those effects anymore. This was revealed in comments made by participants;

"*When I fell sick recently, I was given the yellow and white drug that you take four in the morning, another four in the afternoon and four in the evening. I nearly killed myself because I did not know that the drug was not good. So anytime I take the drug, my entire body begins itching. One day after taking the drug and drinking almost a cup of porridge, I decided to cross the road to buy some stuff, and before I could realise it, I was feeling dizzy and almost knocked down by a car. So I went to the clinic again, and I was told that I should have eaten solid food like rice compared to the porridge, and I should have reported when my body started itching. So from then, I have never taken that drug again.*"

*P6*

"*Oh, at first, when I take the old malaria drugs, the ones you take four tablets at a go, I feel am unable to do anything that very day. But I had malaria recently, and the malaria drugs didn't cause any effect after taking them. The one I took was the one-two tablets a day. I don't remember the names.*"

*P19*

Some participants prefer the antimalarial drugs for the positive side effects they experience when they take medicine.

"*The orthodox drugs like Lonart and Coartem is good; you sweat when you take it when you don't have an appetite it makes you feel like eating, it reduces the fever as well*".

*P8*

## Ineffectiveness of antimalarial drug

When a drug is ineffective, it prevents individuals from patronising it no matter how dire their condition. It also leads most individuals to have less faith in drugs of similar nature. Hence, some participants had a strong feeling of resentment towards using orthodox drugs in treating malaria due to the side effects they experience and how it seems only temporally to cure them of malaria. Also, some participants indicated that they prefer herbal remedies because it work better for them than orthodox drugs are. This was revealed in comments made by participants;

"*For me, I prefer the neem tree leaves treatment because it does not disturb my sleep after taking it, and I can urinate more frequently compared to the hospital drugs, and I also see changes in the colour of my urine. When I take the over the counter orthodox medications, I experience headaches and sometimes loss of appetite for the hospital drugs. The one I got from the pharmacy made me switch to the neem tree leaves because I felt like dying after taking them. I couldn't get up, move about or do anything for that matter, and it didn't even treat malaria*".

*P20*

"*To me, the herbal is excellent better than the orthodox ones, and I prefer it because it gets rid of malaria, no side effects like the orthodox ones and it is cheap too, and you see results an early as possible, but for the orthodox, since it doesn't cure it completely it can lead to other diseases settling in like typhoid.*"

*P12*

## Herbal malaria treatment practices

In Ghana, the herbal treatment of most diseases is highly favoured by most individuals. This is primarily due to Ghanaians' long history of treating conditions with herbs and having a high success rate. This has led to different methods in treating simple diseases like malaria. The participants had other herbal treatment methods for managing malaria. Most of them were more into using the Neem tree and its leaves and using roots and spices mixed in a clay pot called "Odido". Others were also treating malaria with bottled herbal mixtures. This was revealed in comments made by participants;

"*What I know is using the Neem tree and its leaves. You can grind it and make a tea from it and drink it, or you can boil it then pour it into a bucket, add 2 of the big lemons then you sat in front of the bucket and cover yourself with a big cloth which will cover you and bucket from your head to toe, then you inhale the vapour coming from it.*"

*P14*

"*So for the herbal treatment, what I know is that when you buy the mixture like Rooter mixture and Agbeve herbal mixture, they come with a small cup that you can take it three times daily and when you can take one or two bottles, it can treat malaria. So to see that you're getting better, you should at least finish the bottle, and if in a week you can finish one bottle, malaria will be gone. Also, depending on the size of the bottle, you might take a week or two to finish the bottle. Most times after the first day of taking the mixture, you feel a difference in your body which shows the medicine would work; hence you continue to take it.*"

*P9.*

## The effectiveness of herbal medicine use in malaria treatment

The third theme identified the various treatment practices that the participants used in managing malaria. These practices ranged from the use of the Neem tree and its leaves, mixtures, the use of limes and gloves. Some of the participants also used various trees and plants. Three sub-themes emerged from this theme;

## Home preparation for malaria

Most African countries, including Ghana, are heavily influenced by traditional medicine. Before considering alternative medication, most individuals would prepare and try traditional homemade remedies for curative purposes. The participants had so many methods of treating malaria at home. Some first try orthodox medicine, but they switch to herbal ones due to no results. Most of the participants used the Neem tree leaves by boiling them to do steaming, and others too dried the leaves and made tea from them. Some participants also used the mixture called "Odido", and the rest also had some unique combination in which they operated. This was revealed in comments made by participants;

"*So recently, I got malaria and went to the pharmacy. They checked them for malaria parasites, and I was given medications to take. I followed the drug instructions, and after completing it, I was still not better, so I switched to herbal ones, and within three days of using these neem tree leaves, I saw a difference in my health. I cook them, pour them into a bucket, cover myself with cloth and inhale the heat; I sit under it for up to 10–20 minutes, then come out all sweating. I then boil some leaves and drink them without sugar or milk. Surprisingly just three days, I became so fit because when I took medicine I was given at the drug store, I couldn't walk most of the time.*"

*P14*

"*As already said, when I wake up in the morning, I pluck some bitter leaves, wash them with water first, then again wash them with salty water for the second time. I then mix with both hands in a bowl, squeezing fluids out of the leaves and leaving the surface foamy. I sometimes boil water, mix it and then sieve the liquid out; I then add lime and drink. After taking it in, I sometimes visit the washroom so many time., Also, I boil it in hot water with the neem tree for some time. I fetch some down; then I pour the remaining into a bucket cover my head with a*"

*towel with my head facing down into the bucket for some time. After I took the towel off, you will see me all sweaty. I then took the one in the cup I fetched earlier and drank it after"*

**P6.**

Some participants also use pawpaw leaves and fruits like pineapple to make a mixture that is also very effective against malaria.

*"For this mixture, you've got the pineapple in there, Momordica foetida ('ny3nya') and paw-paw leaves. You also add the unripe pawpaw fruit by not peeling it but cutting it into pieces and removing the seeds. You add it to your things, then boil; it's a good medication for malaria".*

**P18.**

### Dosage for herbal preparation

The orthodox medications have a specific, general and fixed dosage; however, the herbal preparations have a fluent dosage system. The dose for that remedy is determined by the particular individual making the remedy or taking the medication and not a standardized one. This was observed as all the participants that took the herbal remedy had their way of taking it, as seen below;

*"For the neem tree mixtures, you can pour from the pot into a gallon to persevere it better. You drink it morning, afternoon and evening. You can take three spoons or just half a cup. When you take for like three days, you should start seeing improvements in your health."*

**P10**

*"So for the bitter leases too, after you add the rock-like salt to it for it to dissolve, then you separate the leaves from it and drink one small cup in the morning and evening till you feel better.".*

**P13**

*"What I know is that for the "Odido" mixture, you can drink it at any time. Depending on how severe your malaria is, you can take it three times a day or twice daily. You can also take it once a day even when you don't have malaria to prevent you from getting malaria."*

**P20**

### Effectiveness of herbal malaria treatment

Ghanaians have a strong sense of acceptance of herbal or traditional remedies in treating diseases; hence, even before using the treatment, they already have a positive perception of how potent or effective it will be. Therefore, almost all the participants that used herbal remedies in treating malaria had a positive attitude towards it due to how effective the remedies were. Most participants stated that they would take herbal treatment for malaria over the orthodox ones anytime because they had no side effects. They see improvements in the shortest possible time. It improves their appetite, and the orthodox ones too are expensive. This was revealed in comments made by participants;

"*For me, I see the herbal remedies to be more effective. Remember I told you I felt sick and tested, to which I was told I had malaria, so when I took the hospital drugs, I was not getting better. When I walk under the sun, then I feel like falling, but when I did the herbal medicine, which was the bitter leaves mixture, I became fit and was moving about malaria-free after testing again at the hospital.*"

*P10*

"*The herbal remedies do so many things in my system, I can't even tell the last time I contracted malaria, yeah, because I don't wait till I contract it before I use the medication, so you treat yourself not only when you get the disease but to prevent it so to me it's very effective. Assuming I have malaria and take it, I start feeling okay at most by four days. This is because the leaves contain no side effects, in the sense that God creates it, and so it is natural.*"

*P7*

"*The local ones are very effective; it's very effective because when I take it on the very first day, I see a difference. After all, if it was my head that was aching me or a fever that I was feeling, I see it to be reduced, if not gone entirely. Still, for the orthodox ones, I would take for a day, three days, and even a week, and I still feel the same was I did before taking it and also one funny thing is the works of the local one on malaria at its core so gets rid of it, but the orthodox does nothing. After taking it, malaria would return in a month or two, and you begin to feel unwell all over again. It's also expensive compared to the herbal ones because the herbal ones are very cheap, even the ones that have been processed.*"

*P12*

## Discussions

Ghanaians are different from other African countries but have one thing in common: how culture influences many aspects of their lives from their music dance, and in respect to the current study, their treatment choice of an illness. The dynamic nature of culture has a significant role in this choice; however, the citizens of most African countries usually settle for herbal medicine for their illnesses even before they consider other treatment options. However, participants indicated that they were not influenced by any cultural practice to resort to the herbal treatment of malaria. However, a study in Nigeria contrasted these results. The authors stated that cultural practices likely to influence appropriate treatment-seeking include cultural perception of malaria as ordinary fever and wrong perceptions of severe malaria attributing it to spiritual means [22].

Influence from family was influential in herbal treatment preference for malaria. This is because it was the only treatment method they observed whilst growing up, hence learning these practices from their parents. However, some of the participants strayed from this path as they grew. This aligned with the findings of other researchers, which revealed that some practices in the family could be passed unto the next younger generations [23].

Another factor that caused participants to use herbal treatment was the anti-malaria side effects. Every drug has its side effects, including antimalarial drugs listed mainly by manufacturers and how to manage them. The current study participant reported having experienced side effects like nausea and vomiting, weakness, and itchiness when taking conventional drugs such as *Coartem or Lonart* hence becoming demotivated from using these drugs. This could be because patients are mostly not informed about the side effects of drugs and how to manage them during prescription. Similar to the current study findings, some authors in Ghana

revealed that their participants taking *Artesunate-amodiaquine (AS+AQ) and Artemether-lumefantrine (AM-L)* had an incidence of adverse events, such as pruritus, fatigue and neutropenia [24]. This is an indication that Ghanaians should be educated on both orthodox and herbal treatment of malaria and its side effect, to help clarify any culture and religious misconceptions about orthodox use to ensure adherence to orthodox herbal treatment, and effective management and side effect.

A perception that orthodox medication was not effective also influenced their preferences for herbal medicine for treating malaria. Most patients in this study felt disappointed when their symptoms were not relieved after taking anti-malaria drugs prescribed at the hospital and resorted to herbal treatment. However, the study of some researchers highly disagrees with the current study results as the use of artemisinin-based combination therapy (ACT) with a competent partner drug and having multiple ACT as a first-line treatment choice for malaria sustained high levels of effectiveness [25]. This is further illustrated with the global effectiveness of artemisinin-based drugs being 67.4% (IQR: 33.3–75.8), 70.1% (43.6–76.0) and 71.8% (46.9–76.4) for the 1991–2000, 2006–2010, and 2016–2019 periods, respectively. Hence, the ineffectiveness reported in the current study may be influenced by several factors, such as non-adherence. This perception may be influenced by cultural beliefs that herbal medicines work better and could only be rectify through increase health promotion and intensification of education of orthodox treatment of malaria on the various social media platforms.

Ghana, being majorly populated by plants and vegetation, availed most Ghanaians to use leaves, herbs, spices, and parts of various plants to treat most of their illness. They could develop herbal remedies for most of the common conditions that affected them the most through trial and error. Malaria is one of the prevailing conditions in the country, has a lot of herbal treatment practices in curing it, and the participants of this study had various ways of treating it. Most of them used the neem tree for their remedies, from steaming inhalation with the boiled leaves, drying the leaves, and making a tea solution. Others, too, would buy the bottled herbal mixtures sold to treat malaria, which contain different sorts of plants. Similar to the findings is a study by authors who revealed 29 species of plants belonging to 22 families being sold for the treatment of malaria [5].

The most frequently mentioned species of plants were *Morinda Lucida Benth.*, *Indigofera sp*. and *Nauclea latifolia Sm*. The majority (82.8%) of the plant materials were sold in the dried state, and 6.9% were sold in a new form. Also comparable to the above results is the study that ascertained that twenty-seven species were used to treat malaria [26]. The most frequently mentioned were *Vernonia amygdalina*, *Momordica foetida*, *Zanthoxylum chalybeum*, *Lantana Camara and Mangifera indica*. Drugs from these plants were prepared from single species as water extracts and were administered in variable doses over varied periods.

The majority of the participants combined various leaves, herbs, and different parts of multiple plants to prepare the herbal remedy they used in treating malaria. The participants' most common preparation was using the *neem* tree leaves by boiling them and inhaling the vapor under a cloth. Also, some would mix salt with bitter leaves and drink the solution from that. Other participants also burned pineapple, *Momordica foetida*, and pawpaw leaves and consumed the preparation for malaria treatment, and to them, it's very good malaria. A study in Nigeria had a similar finding to the current research. The authors stated that forty-one plant species belonging to 27 families were identified as being used locally for antimalarial herbal recipes [27]. Of these, *Enantia chlorantha (31.5%)*, *Carica papaya (27.5%)*, *Azadirachta indica (25.5%)*, *Cymbopogon citrates (23.3%)*, *Morinda Lucida (22.7%)*, *Mangifera indica (21.1%)*, and *Alstonia boonei* (20.5%) were the most frequently used plants.

Unlike the hospital drugs that come with a standardized prescription and dosage, the herbal preparation is mainly left to the discretion of the individual preparing it. The individual

preparing the herbal remedy focuses on severity and experience to give a dosage for the treatment. This is likely to predispose herbal drug users to some severe complications. Compared to the hospital drugs, the herbal preparation does not have any system for the prescription and dosage, which can be very dangerous. Still, as the current study participants demonstrate, they are healed following their dosage. A study had comparable results to the present study as traditional healers treat malaria and a wide range of other health problems using medicinal plants of unverified efficacy in various unstandardized dosage forms [28]. Hence, the authors stated that there is a need for scientific evaluation and standardization of these formulations and dosages, if found effective, to eliminate the possibility of short to long term toxic effects.

Most of the study participants stated that they use either a spoon or a cup to measure the quantity of the remedy they should take. Most participants said that they could take half a cup three times in a day or three spoons in the morning, afternoon, and evening. The participants did not have a duration for taking the remedies but most stated that usually, after three days, they get better and stop taking the pills. Agreeing with the above results is a study whose authors noted that a large majority of respondents in all the ethnic groups claimed to use the same herbs for the treatment and prevention of malaria, and significant improvement is experienced after use [29]. There is usually no specific dose or dose regimen; however, a high proportion in all the ethnic groups use herbal preparation thrice a day, and a few of the respondents take unspecified measures at arbitrary intervals.

The effectiveness of a drug is a crucial component in boosting the patient's trust in the drug. This effectiveness is best seen when a patient experiences no adverse side effects from the medication, and the drug also performs its desired effect. The study participants shared a similar belief in herbal remedies for malaria treatment as the majority of them stated that herbal remedies are very effective against malaria. Compared to the hospital drugs that can even worsen malaria, the herbal remedies eradicate malaria at its core hence effectively destroying the malaria parasite. Similar to the above findings is a study whose authors stated that the resolution of the cardinal symptoms was also observed in most participants by day 7 [4]. The treatment also had a good safety profile as none of the participants reported any adverse effects. Liver, kidney, and hematological profiles were also customary after the study. Also, a study indicated that herbal treatment of malaria is very effective as the majority of the respondents had faith in the traditional medicine of malaria [30].

## Conclusion

This study discovered that several factors influenced participants' preferences for malaria treatment. Participants further listed some traditional ways of treating malaria which implies that there is herbal malaria practice. However, literature in this area is inadequate, and most herbs lack specifications for use. It is therefore recommended that future research focus on scientific herbal malaria treatment. Also, regulating bodies should ensure that quality herbal drugs are sold for consumption.

## Acknowledgments

The researchers want to express their gratitude to all authors whose work was cited in this study and the study participants.

## Author Contributions

**Conceptualization:** Evans Osei Appiah, Albert Oppong-Besse, Priscilla Ofosuhemaa Asiedu.

**Data curation:** Evans Osei Appiah, Albert Oppong-Besse, Priscilla Ofosuhemaa Asiedu.

**Formal analysis:** Evans Osei Appiah, Albert Oppong-Besse, Priscilla Ofosuhemaa Asiedu.

**Investigation:** Evans Osei Appiah, Stella Appiah, Ezekiel Oti-Boadi, Albert Oppong-Besse, Dorothy Baffour Awuah, Lt Emmanuel Oti-Boateng.

**Methodology:** Evans Osei Appiah, Stella Appiah, Ezekiel Oti-Boadi, Albert Oppong-Besse, Dorothy Baffour Awuah, Lt Emmanuel Oti-Boateng.

**Project administration:** Ezekiel Oti-Boadi.

**Resources:** Evans Osei Appiah, Ezekiel Oti-Boadi, Albert Oppong-Besse, Dorothy Baffour Awuah, Lt Emmanuel Oti-Boateng.

**Software:** Evans Osei Appiah, Stella Appiah, Ezekiel Oti-Boadi, Albert Oppong-Besse, Dorothy Baffour Awuah, Lt Emmanuel Oti-Boateng.

**Supervision:** Evans Osei Appiah.

**Validation:** Evans Osei Appiah, Stella Appiah, Ezekiel Oti-Boadi, Dorothy Baffour Awuah, Priscilla Ofosuhemaa Asiedu, Lt Emmanuel Oti-Boateng.

**Visualization:** Evans Osei Appiah, Stella Appiah, Ezekiel Oti-Boadi, Dorothy Baffour Awuah, Priscilla Ofosuhemaa Asiedu, Lt Emmanuel Oti-Boateng.

**Writing – original draft:** Evans Osei Appiah, Stella Appiah, Ezekiel Oti-Boadi, Albert Oppong-Besse, Dorothy Baffour Awuah, Priscilla Ofosuhemaa Asiedu, Lt Emmanuel Oti-Boateng.

**Writing – review & editing:** Evans Osei Appiah, Stella Appiah, Ezekiel Oti-Boadi, Albert Oppong-Besse, Dorothy Baffour Awuah, Priscilla Ofosuhemaa Asiedu, Lt Emmanuel Oti-Boateng.

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
