## [Decision Letter · Decision Letter 0]

23 May 2022

PONE-D-22-06121PRACTICES OF HERBAL MANAGEMENT OF MALARIA AMONG TRADING MOTHERS IN SHAI OSUDOKU DISTRICT, ACCRA.PLOS ONE

Dear Mr Evans Osei Appiah

Thank you for submitting your manuscript to PLOS ONE. After careful consideration, we feel that it has merit but does not fully meet PLOS ONE’s publication criteria as it currently stands. Therefore, we invite you to submit a revised version of the manuscript that addresses the points raised during the review process.

We look forward to receiving your revised manuscript.

Kind regards,

Michael Ansong (PhD)

Academic Editor

PLOS ONE

Journal Requirements:

2. We note you have included a table to which you do not refer in the text of your manuscript. Please ensure that you refer to Tables 1 and 2  in your text; if accepted, production will need this reference to link the reader to the Table.

Additional Editor Comments (if provided):

Malaria is undoubtedly one of the leading causes of morbidity in the world, including in the country where the study was conducted. In the paper, the authors explored the practices of herbal management of malaria among trading mothers. Qualitative methodology was used.

I have a few comments that I would like the authors to address as they revised the manuscript.

1. Could there be a clear clarification and justification for why Trading mothers were used? And why the study area was selected among numerous markets in Ghana?

2. It is well established that there are four aspects of trustworthiness that qualitative researchers must establish: credibility, dependability, transferability, and confirmability. In the current study, the authors tried to explain how credibility was established ie explained as the accuracy and authenticity of a research finding. Credibility essentially asks the authors to clearly link their findings with reality in order to demonstrate the truth of the research study’s findings. This is not well clarified in the current version of the paper. Also, the other aspect of trustworthiness ie dependability, transferability and confirmability must clearly be explained in the paper.

3. The authors should be specific about which approach of content analysis did they use: conventional, directed, or summative?

4. Result section, the second line … “The participants were between twenty-six (26) and fifty-nine (59)”?? Sentence is not complete.

5. What does JSS, SSS and Form 4 level of education means?

6. Table 1..The Title should be made complete

7. Table 2…Table should be well formatted..lines are missing

8. Overall, the paper needs to be well formatted, for example, the section titles and Tables should follow the journal style. All the Tables should be cited in the text.

Reviewers' comments:

Reviewer's Responses to Questions

**Comments to the Author**

1. Is the manuscript technically sound, and do the data support the conclusions?

Reviewer #1: Partly

2. Has the statistical analysis been performed appropriately and rigorously? 

Reviewer #1: N/A

3. Have the authors made all data underlying the findings in their manuscript fully available?

Reviewer #1: Yes

4. Is the manuscript presented in an intelligible fashion and written in standard English?

Reviewer #1: Yes

5. Review Comments to the Author

Reviewer #1: The conclusion does not reflect or emanate from the present study. The current conclusion looks more of a literature review. Authors should summarize their key findings and implications of the findings.

6. PLOS authors have the option to publish the peer review history of their article (what does this mean?). If published, this will include your full peer review and any attached files.

Reviewer #1: No

---

## [Author Response · Author response to Decision Letter 0]

26 May 2022

REVIEWER COMMENT RESPONSE

EDITOR We note you have included a table to which you do not refer in the text of your manuscript. Please ensure that you refer to Tables 1 and 2 in your text; if accepted, production will need this reference to link the reader to the Table. Table 1 and 2 have been included in the test as suggested

EDITOR 3. In your Data Availability statement, you have not specified where the minimal data set underlying the results described in your manuscript can be found. PLOS defines a study's minimal data set as the underlying data used to reach the conclusions drawn in the manuscript and any additional data required to replicate the reported study findings in their entirety. All PLOS journals require that the minimal data set be made fully available. For more information about our data policy, please see http://journals.plos.org/plosone/s/data-availability. All data are made available and attached as supplementary files

EDITOR 4. Please include your full ethics statement in the ‘Methods’ section of your manuscript file. In your statement, please include the full name of the IRB or ethics committee who approved or waived your study, as well as whether or not you obtained informed written or verbal consent. If consent was waived for your study, please include this information in your statement as well.

 Revision has been made as suggested

REVIEWER 1 1. Could there be a clear clarification and justification for why Trading mothers were used? And why the study area was selected among numerous markets in Ghana? These trading mothers were selected from rural communities where malaria was prevalent and mothers who came to trade were from different communities in the district

REVIEWER 1 2. It is well established that there are four aspects of trustworthiness that qualitative researchers must establish: credibility, dependability, transferability, and confirmability. In the current study, the authors tried to explain how credibility was established ie explained as the accuracy and authenticity of a research finding. Credibility essentially asks the authors to clearly link their findings with reality in order to demonstrate the truth of the research study’s findings. This is not well clarified in the current version of the paper. Also, the other aspect of trustworthiness ie dependability, transferability and confirmability must clearly be explained in the paper. This has been revised and shown in track changes

REVIEWER 1 3. The authors should be specific about which approach of content analysis did they use: conventional, directed, or summative? Conventional approach was used and has been included as suggested

REVIEWER 4. Result section, the second line … “The participants were between twenty-six (26) and fifty-nine (59)”?? Sentence is not complete. This has been corrected

REVIEWER 5. What does JSS, SSS and Form 4 level of education means?

6. Table 1..The Title should be made complete

7. Table 2…Table should be well formatted..lines are missing

8. Overall, the paper needs to be well formatted, for example, the section titles and Tables should follow the journal style. All the Tables should be cited in the text. REVISION MADE TO THESE COMMENTS

Reviewer Remove full stop after the title.

Change “the researchers, therefore aim … “ to “ The study aims at …”

The conclusion in the abstract is misplaced. Authors must summarize their findings

Complete the statistics on the number of deaths attributed to malaria. Is it per annum or monthly?

Scientific names of trees should be written in italics. Full stop has been removed

This has been revised

The conclusion has been revised

This has been completed as suggested

Scientific names have been italicized

 Researchers (plural) has been used in the preceding paragraph but researcher (singular) has been used here. The same is repeated in the last paragraph and other pages of the manuscript

Change “a) women who have mentally impaired” to “women who are mentally impaired”.

“ … that were shown” make the sentence confusing.

Capitalise the first letters of Ghana health service. That is “Ghana Health Service”.

What sort of compensation was given to participants? Correct it if it is a mistake.

Change the sentence from future tense to past tense. That is, “ … will be …” to “was”.

What does the figures 26 and 59 stand for? I presume they are age ranges.

Add “of respondents”. That is Demographic characteristics of respondents

First income bracket should be 100-1000 and NOT 100-100.

Add the currency

The content is ambiguous. Revision has been made

This has been changed appropriately

It has been changed to “A tape recorder was used to capture the interviews during the recording

Corrected

This has been corrected

Corrected

Corrections made

Respondents have been added

Corrected

The content has been revised

REVIEWER Change “younger” to “young”. CORRECTED

Authors write “… prefer herbal remedies due to how ineffective the orthodox are.” This statement presumes orthodox medicine are ineffective. This statement is not backed by any scientific evidence. Authors should be mindful that their findings are based on perceptions of respondents. This should reflect in the sentence above. 

It has been revised appropriately as suggested

REVIEWER “…when I take it …” is confusing. What does the “it” refers to? I suggest you replace “it” with “orthodox medicine” and use the pronoun later.

because I felt dying…” should be change to “ … because I felt like dying…”

“ … I can’t get up …” should be change to “…. I couldn’t get up”

Change “sit” to “sat”

Change “… after I take towel off …” to “after I took …”

Change “the dose for that remedy was determined by …” to “… that remedy is determined by …”

 Revised

Revision made

Revised

Revised

REVIEWER Change “… you separate the leaves for it” to “… you separate the leaves from it”

Change “what I know off is that…” to “what I know is that ... ”

Change “ … they would take herbal treatment of malaria over …” to “ they would take herbal treatment for malaria over …”

Changed “ … contracted these results.” to “ … contrasted these results”. This has been changed as suggested

Corrected

Changes have been made

This has been revised

REVIEWER Authors could reflect on the influence of education and religion on the Ghanaian culture. This has been included in the discussion

REVIEWER The conclusion should emanate from the research findings. What authors currently have written are problem statement and recommendations. The conclusion has been revised

---

## [Editor Report · Decision Letter 1]

6 Jul 2022

PRACTICES OF HERBAL MANAGEMENT OF MALARIA AMONG TRADING MOTHERS IN SHAI OSUDOKU DISTRICT, ACCRA

PONE-D-22-06121R1

Dear Mr Appiah

We’re pleased to inform you that your manuscript has been judged scientifically suitable for publication and will be formally accepted for publication once it meets all outstanding technical requirements.

Kind regards,

Michael Ansong, PhD

Academic Editor

PLOS ONE

Additional Editor Comments (optional):

The paper needs to be formatted according to the journal's style.
---

## [Editor Report · Acceptance letter]

8 Jul 2022

PONE-D-22-06121R1 

PRACTICES OF HERBAL MANAGEMENT OF MALARIA AMONG TRADING MOTHERS IN SHAI OSUDOKU DISTRICT, ACCRA 

Dear Dr. Appiah:

I'm pleased to inform you that your manuscript has been deemed suitable for publication in PLOS ONE. Congratulations! Your manuscript is now with our production department. 

Kind regards, 

on behalf of

Dr. Michael Ansong 

Academic Editor

PLOS ONE